# Climate change-induced shifts in the food systems and diet-related non-communicable diseases in sub-Saharan Africa: a scoping review and a conceptual framework

Janet Tapkigen [1], Seeromanie Harding,[2] Jutta Pulkki,[1] Salla Atkins,[1,3]
Meri Koivusalo [1,4]

For numbered affiliations see end of article.

**Correspondence to**
Janet Tapkigen;
janet.tapkigen@tuni.fi

## ABSTRACT

**Objectives** To determine the relationship between climate change, food systems and diet-related non-communicable diseases (DR-NCDs) in sub-Saharan Africa (SSA) and propose a conceptual framework for food systems in SSA.
**Design** A scoping review.
**Eligibility criteria** Studies included investigated the relationship between climate change and related systemic risks, food systems, DR-NCDs and its risk factors in SSA. Studies focusing on the association between climate change and DR-NCDs unrelated to food systems, such as social inequalities, were excluded.
**Sources of evidence** A comprehensive search was conducted in ProQuest (nine databases), Google Scholar and PubMed in December 2022.
**Charting methods** Data extracted from studies included author, study type, country of study, climate change component, DR-NCD outcomes and risk factors, and impacts of climate change on DR-NCDs. A narrative approach was used to analyse the data. Based on the evidence gathered from SSA, we modified an existing food system conceptual framework.
**Results** The search retrieved 19 125 studies, 10 of which were included in the review. Most studies used a cross-sectional design (n=8). Four explored the influence of temperature on liver cancer through food storage while four explored the influence of temperature and rainfall on diabetes and obesity through food production. Cross-sectional evidence suggested that temperature is associated with liver cancer and rainfall with diabetes.
**Conclusion** The review highlights the vulnerability of SSA's food systems to climate change-induced fluctuations, which in turn affect dietary patterns and DR-NCD outcomes. The evidence is scarce and concentrates mostly on the health effects of temperature through food storage. It proposes a conceptual framework to guide future research addressing climate change and DR-NCDs in SSA.

## INTRODUCTION

Non-communicable diseases (NCDs) account for 74% (41 million) of all deaths globally every year.[1] The primary causes

## STRENGTHS AND LIMITATIONS OF THIS STUDY

⇒ This scoping review followed the Joanna Briggs Institute methodology for scoping reviews.
⇒ The methodology and search terms involved an extensive search of literature.
⇒ Existing food system framework conceptually guided the methodology and development of hypotheses.
⇒ Despite efforts to search for grey literature, country-specific documents may have been missed; this study is based on published data and, therefore, is subject to publication bias.
⇒ The studies that met the inclusion criterion were primarily observational and descriptive; thus, while these provided valuable insights for hypotheses, their contribution to evidence regarding causality is limited.

of NCD-related deaths are cardiovascular diseases (accounting for 17.9 million deaths annually), cancer (9.3 million), chronic respiratory diseases (4.1 million) and diabetes (2.0 million).[1] Collectively, these diseases contribute to 80% of all NCD-related deaths.[1] In sub-Saharan Africa (SSA), cardiovascular diseases are the leading level 2 causes of NCD-related deaths, accounting for 15.1% of the NCD burden. Diabetes and liver cancer also make significant contributions to the NCD burden in the region, accounting for 4.2% and 0.8% of NCD-related deaths, respectively.[2] These three diseases—cardiovascular diseases, liver cancer and diabetes—are known as diet-related NCDs (DR-NCDs), as they are primarily caused by poor dietary habits, characterised by highly processed foods high in saturated fats, sugars and salt.[3]

To effectively address the burden of DR-NCDs in SSA, it is essential to understand how the food systems function, what their drivers are and their role in shaping

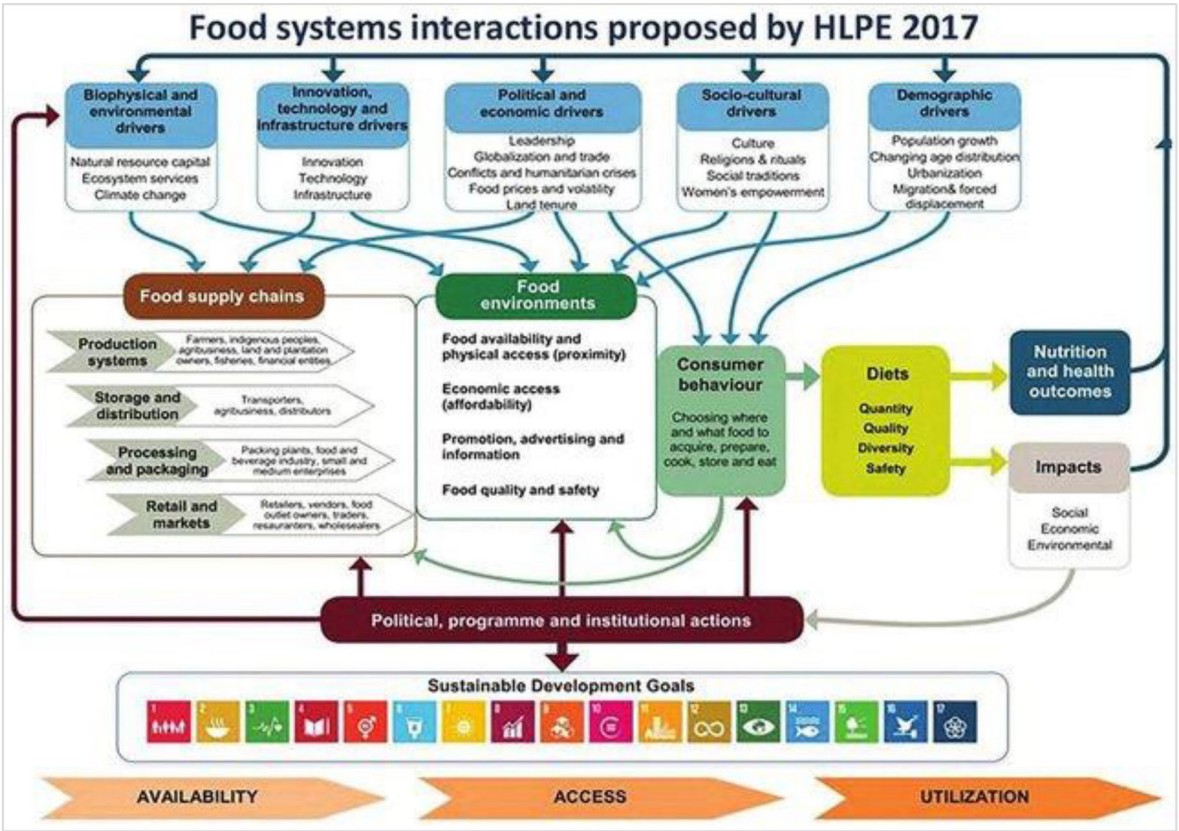

**Figure 1** HLPE. Nutrition and food systems. A report by the High Level Panel of Experts on Food Security and Nutrition of the Committee on World Food Security, Rome. 2017 [18] . This is the original HLPE framework. The food supply chains in this framework are henceforth referred to as pathways in this review. HLPE- high-level panel of experts.

diets. Food systems can be described as interconnected networks comprising the environment, people, processes, infrastructure and institutions involved in various aspects of food production, processing, transportation, storage, distribution, retailing and consumption.[4] The outcomes of these activities and their governance collectively define the functioning of the food system.[4] The interconnected components of food systems, namely the food supply chain, food environments and consumer behaviour, are illustrated in figure 1.

The food supply chains encompass all stages from production to waste disposal,[5] and they depend on ecological, human, energy and economic systems to produce and distribute foods.[6] On the other hand, food environments encompass the settings where consumers make food-related choices,[5] therefore, connecting food supply and demand.[6] Food environments are divided into personal environments (accessibility, affordability, convenience, desirability) and external environments (food vendors, prices, income levels, knowledge, sociocultural norms, marketing practices, regulations).[7] For certain individuals, these environments are local (involving self-produced or locally sourced foods), while for others, they are global (encompassing foods from regional and international sources).[5]

Food environments influence consumer behaviour,[6] which refers to the decisions and actions individuals take when acquiring and consuming food.[4] These decisions shape diets in terms of quantity, quality, diversity, safety and adequacy of food.[6] Consequently, these diets exert influence on interconnected systems, including nutrition outcomes that directly impact healthcare systems and the climate impacts of diets that affect ecosystems.[6]

SSA is currently facing a food systems challenge related to climate change, poor agricultural practices, food insecurity and the consequences of DR-NCD risk. Climate change threatens the region's ecosystems, agricultural productivity and food security, exacerbating existing vulnerabilities. Climate change, driven by human activities, has been widely acknowledged as a significant contributor to the development and exacerbation of DR-NCDs.[8] The rising global temperatures, erratic rainfall patterns and changing patterns of extreme weather events significantly influence agricultural and food systems.[9] These climatic shifts impact food quality, availability and affordability,[9 10] potentially altering food access and choices, with implications for nutrition and dietary habits.[10] As food systems are undergoing profound transformations driven by population growth, globalisation, technology, urbanisation, trade liberalisation, migration, policies, competition for essential resources (water, energy, land) and shifts in income distribution, the impacts of climate change compound the effects of these drivers.[4] As such, climate

change can be seen as amplifying the underlying drivers associated with DR-NCDs.[8 9]

To effectively address the burden of DR-NCDs in SSA amid these climate-induced changes, a food systems approach is necessary.[11] Unlike disease-oriented approaches, food systems approach consider the entire food system—from production to consumption.[12] Existing interventions for NCDs in SSA have not focused on multisectoral pathways on determinants of health.[13 14] Health promotion activities to prevent NCDs have focused on individual responsibility and influencing purchase patterns more than a systems approach where the type and availability of food itself are considered. Therefore, shifting towards a food systems approach allows for a holistic and comprehensive understanding of the entire food system, recognising the intricate interplay and feedback loops between different risk factors for NCDs.[15]

In the context of addressing DR-NCDs, adopting a systems thinking approach acknowledges that one system alone, for example, the healthcare system, cannot solely determine health outcomes.[16] Instead, it recognises the significant role interconnected systems (such as the environment, energy and government sectors) play in shaping environments that impact DR-NCDs.[16] However, implementing a food systems approach in SSA comes with challenges resulting from underreporting underlying determinants of DR-NCDs[17] and limited evidence on the specific pathways in which climate change and related systemic risks influence food systems and, in turn, on DR-NCDs in the region.

Existing food system frameworks have explored the broad-scale interactions and pathways in which climate change influences the risks associated with nutrition outcomes. For instance, the high-level panel of experts (HLPE)[18] developed a food systems framework in 2017, illustrating the interplay between diverse drivers within food systems, including climate change and their effects on health and nutrition outcomes. The HLPE, formed in 2009, is a United Nations body responsible for assessing food security and nutrition research.[19] HLPE is governed by 15 world-renowned scientists who are experts in food security and nutrition, and the aim is to assess the current state of food security and nutrition and its underlying causes.[19]

While the HLPE framework[18] and previous evidence[8 20] have provided valuable broad-scale perspectives, there is an urgent need to compile, synthesise and contextualise more localised and region-specific research and to develop contextualised frameworks that can guide future studies. We, therefore, used the original HLPE framework (figure 1) and existing literature to hypothesise and synthesise the causal pathways in which climate change can impact DR-NCDs in SSA.

## Aims of the review
This review aims to explore and synthesise research originating from SSA on climate change impacts on food systems and DR-NCDs and to examine if we need to complement and amend the HLPE framework to the SSA context.

## METHODS
### Conceptual framework development
This study used the original HLPE food systems conceptual framework[18] to hypothesise pathways through which climate change affects food systems and, in turn, DR-NCDs in SSA. The framework is presented in figure 1 and includes four food supply chain domains: production systems, storage and distribution, processing and packaging, and retail and markets. These supply chains are henceforth referred to as pathways in this review.

In this review, climate change-related systemic risks include any impact of climate change events on other sectors or systems that subsequently influence food systems.

Based on the HLPE framework[18] and knowledge drawn from scientific literature,[21–26] we hypothesised pathways in which climate change may affect DR-NCDs and its risk factors (ie, obesity, overweight) in SSA. These pathways include:

### The food production pathway
Regarding the food production pathway, we hypothesise that the effects of climate change could reduce crop production and yields, consequently reducing food availability. Reduction in food availability could increase food prices, influencing food affordability and, in turn, impact health outcomes.[22]

### The food storage pathway
Under the food storage pathway, we propose that reliance on stored crops vulnerable to the impacts of extreme rainfall events can have adverse effects on both the quantity and quality of food. Wet conditions can create a conducive environment for the growth of infectious pathogens. These pathogens, affecting crops during harvesting and storage, could lead to the production of carcinogenic aflatoxins (AF) produced by certain moulds, which have been associated with hepatocellular carcinoma, a primary contributor to liver cancer.[26]

### The food distribution and market pathway, and the food processing and packaging pathway
Within the contexts of food distribution and market pathways and food processing and packaging pathways, we hypothesise that extreme weather events, such as floods, might disrupt distribution networks, infrastructure and trade channels, resulting in food losses—especially perishable goods.[21 25] These disruptions can impact the affordability and diversity of food items, impacting food and nutrition security. When comparing fresh perishable foods to processed alternatives, processed foods tend to withstand environmental impacts more effectively, making them a preferable choice within climate variability, although with potential trade-offs concerning

nutrition value and health implications.[21 25] Moreover, decreased food availability can increase food prices, potentially creating a scenario where imported goods become more easily accessible. Often, these imported products lean towards cost-effectiveness and a high degree of processing.

### The food retail pathway
In the context of the food retail pathway, we hypothesise that extreme weather patterns could reduce crop production and disrupt supply chains for food retailers. Reduced food production and disruptions in the supply chains could lead to shortages of certain food items, affecting consumer choices and potentially changing dietary patterns. For instance, if fresh produce becomes scarce due to climate-related disruptions in production and distribution, consumers may opt for less healthy processed alternatives.

### The food consumption pathway
In the food consumption pathway, we posit that climate variability may influence consumption patterns by affecting availability, affordability and acceptability.[27] These shifts could influence food quantity, quality, and prices and reduce income.[27] These combined effects can have multifaceted implications for nutritional access, availability and affordability. This can lead to shifts in dietary preferences, often towards more economical yet highly processed food options.[23] The health impact of this pathway is likely to be more important in the light of the focus on processed foods in practice.

## METHODS
### Study design
To augment the HLPE framework with the evidence from the region and to build a context-relevant framework (HLPE-SSA), we conducted a scoping review following the methodological guidance by Joanna Briggs Institute (JBI).[28] The method includes (1) identifying the research question; (2) inclusion criteria; (3) concept; (4) context; (5) types of evidence sources; (6) search strategy; (7) evidence screening and selection; (8) data extraction; (9) data analysis and (10) presentation of results.

Scoping reviews are helpful when the literature is complex and serve as a tool to provide policy-makers with a comprehensive understanding of the nature of the issue and how that concept has been studied within the existing body of literature.[28] This review was reported per the Preferred Reporting Items for Systematic Reviews and Meta-analysis reporting guidelines.[29] This study was based on a registered protocol (https://osf.io/9unbd).

### Research question
What kind of research-based evidence exists on climate change, food systems and DR-NCDs relationship from SSA?

### Inclusion criteria
We included studies that:
1. Investigated the relationship between climate change and related systemic risks, food systems, and DR-NCDs and their risk factors.
2. Were available as full texts.
3. Had an English abstract or title.
4. Were related to/conducted in SSA.
   Studies were excluded if they:
1. Examined the association between climate change and DR-NCDs through pathways that do not include food systems, such as social inequalities.
2. Were not available as full texts.
3. Were not conducted in or related to SSA.
4. Were editorials, letters to the editor or theoretical discussions of potential impacts or planned programmes that have not yet been implemented, theses, newspapers, magazines, conference abstracts, podcasts, radio programmes and blogs.

### Concept
This review primarily focused on investigating the relationship between climate change and interconnected systemic factors (eg, urbanisation, globalisation, migration and food prices), on food systems and subsequently on DR-NCDs and its risk factors.

### Context
The scope of this review was centred on examining studies carried out specifically in SSA. These countries are listed in online supplemental table 1. Online supplemental table 1 includes the names of SSA countries, population, GDP per capita and income category, all retrieved from the World Bank website.[30]

### Search strategy
The pilot search of PubMed enabled the identification of key terms for the search strategy. The search terms were developed guided by JBI's 'Population, Concept and Context' framework and tested in an iterative process to ensure relevance. It included three components: (1) climate change*, (2) SSA countries, (3) DR-NCDs* and nutrition outcomes*. The detailed search strategy is available in online supplemental appendix 1. The identified search terms were adapted for other databases. No language restriction was applied in the search. The search started from 1995, when the Global Framework for Climate Change was instituted to address the challenges posed by global warming.[31 32] We included studies published in peer-reviewed journals between 1995 and 2022.

### Grey literature search strategy
The search strategy for grey literature was adapted for each data source. The search terms are presented in online supplemental appendix 2. Grey literature published from 1995 was considered.

## Types of evidence sources

Searches were conducted in ProQuest-9 databases (ProQuest Central, Health and Medical Collection, Environment Science Database, Public Health Collection, Research Library: Health and Medicine, Middle East and Africa Database, Social Science Database, Social Science Premium Collection, and Agriculture Science Database) and PubMed in December 2022. Reference list of relevant citations was further manually searched to identify any articles of interest.

### Grey literature data sources

The peer-reviewed searches were supplemented by Google Scholar search in July 2023 involving the first 20 pages. Subsequently, in February 2024, another search was conducted in Google Scholar expanding the search to the first 80 pages as suggested by Haddaway *et al*[33] to gain further reach for grey literature. Published and unpublished reports, policy documents and relevant materials from government departments, agencies and other organisations were also searched for in Google searches, and targeted websites including International Food Policy Research Institute, Consultative Group on International Agricultural Research, WHO, Food and Agriculture Organization, World Food Programme and World Bank in July 2023. Additionally, a sample of seven government websites in SSA was searched for in February 2024. These seven countries (Kenya, Malawi, Nigeria, Zambia, Mauritius, South Africa and Botswana) were chosen based on recently reported climate change related disasters[34] and to gain coverage from each income group: lower middle income, low income, upper middle income and high income.[35] We searched the publications tab of the Ministry of Health website, Ministry of Agriculture and Environment websites.

### Evidence screening and selection

Following the search, all identified citations were collated and uploaded into Mendeley, where duplicates were removed. The studies were then uploaded to Rayyan.ai[36] for the title and abstract screening. Two authors, JT and MK, independently screened the titles and abstracts. Full texts of relevant studies were assessed for inclusion independently by two reviewers (JT and MK). Disagreements were resolved through discussion.

The search process identified 19 125 studies after removing 1221 duplicates. Among these, 7286 studies were peer-reviewed articles, and 11 239 were grey literature. We excluded 18 428 records based on irrelevant titles and abstracts, leaving us with 106 studies for which we retrieved the full texts for detailed assessment. Of these, 10 studies met the eligibility criteria and were included in this review. No studies from the grey literature met our inclusion criteria. The search process is depicted in figure 2.

### Data extraction

JT extracted quantitative and qualitative data in Excel using an extraction form developed for the present study (online supplemental appendix 3). The data included the following characteristics: author, study type, country of study, climate change component discussed, DR-NCD outcomes and risk factors, and impacts of climate change on DR-NCDs.

### Data analysis

A narrative synthesis[37] approach was applied in this review and presented in tables where appropriate. This involved integrating quantitative and qualitative studies through collation, analysis, synthesis and presentation.[37]

### Patient or public involvement

None.

## RESULTS

The results are presented under three main sections: (1) characteristics of included studies; (2) results on the influence of climate change on DR-NCDs and (3) a conceptual framework for food systems and DR-NCDs in SSA. The results are presented according to the identified food system pathways.

### Characteristics of included studies

Ten studies met our inclusion criteria,[38–47] including eight quantitative[38 40–44 46 47] and two qualitative studies.[39 45] The majority of studies used cross-sectional design (n=8).[38 40–44 46 47] The sample size ranged from 28 to 40 300.

Seven studies were published in the 2020s,[38–42 44 45] two in the 2010s[43 47] and one in the 1990s.[46] Three studies were conducted in Kenya,[39 40 47] two in Ghana,[42 45] with one study each in Cameroon,[43] Malawi,[44] Sudan,[46] Tanzania[41] and Zimbabwe,[38] as shown in table 1.

Among the studies included, four studies[38 42 43 47] used primary data collection methods to estimate DR-NCD outcomes and its risk factors. Three studies[40 41 44] drew on data from national health sources to recruit participants. All three studies,[40 41 44] using secondary data, relied on original survey data to measure and estimate health outcomes.

Three studies[38 42 47] used AF and FM levels from grain and milk samples to quantify dietary exposure and estimate the risk of liver cancer. Meanwhile, five studies[40 41 43 44 46] used human samples to assess dietary exposure and DR-NCD risk. Notably, two studies[39 45] took an alternate approach, looking at participants' perceptions to illustrate health risks associated with climate change.

The most common climate variable investigated was temperature (related to increased humidity), reported in 5 out of 10 studies.[38 41 42 46 47] In this review, if a study examined the association between temperature and subsequent rise in humidity levels (above 14% moisture content), we classified the climate variable as temperature (related to increased humidity). Four studies investigated

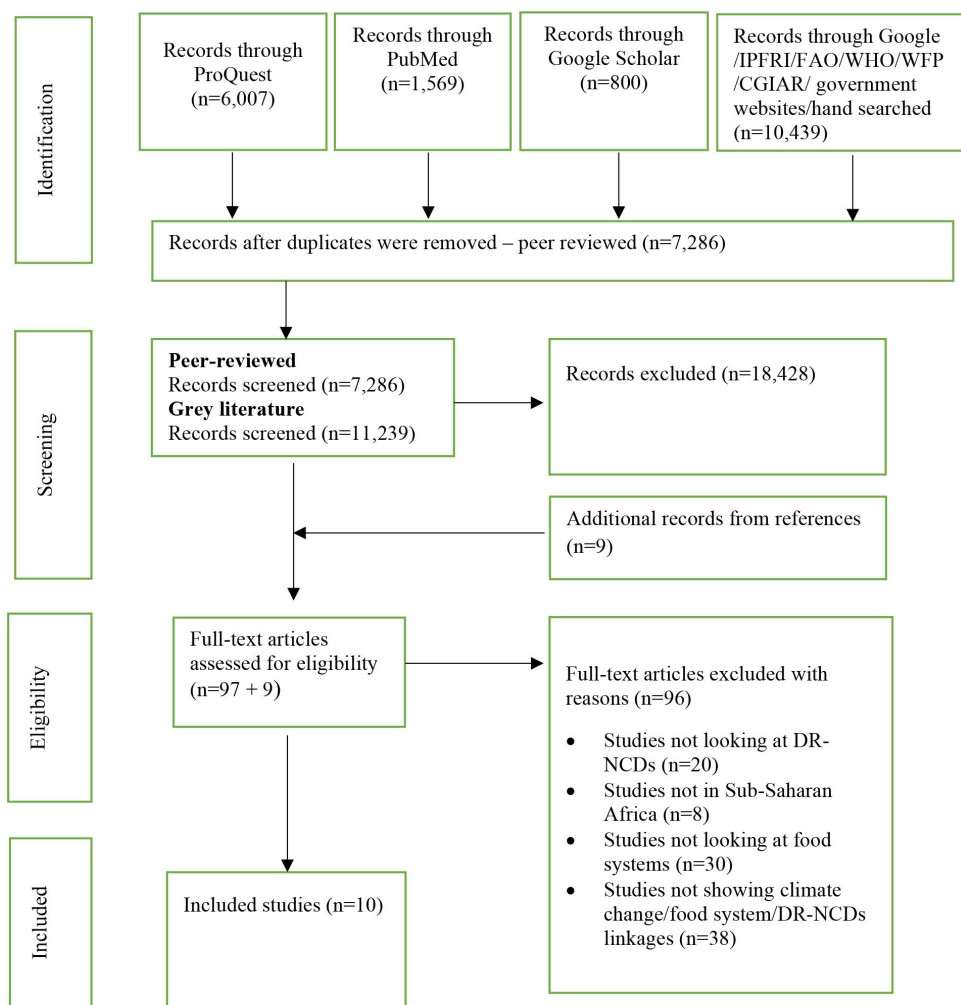

**Figure 2** PRISMA flow diagram: This diagram shows the systematic process we followed to include articles by our search framework.

DR-NCD outcomes related to variation in rainfall[40 43 44 47] and temperature.[40 41 43 47]

Five cross-sectional studies measured[41 46] and estimated[38 42 47] liver cancer rates. Two studies looked at the relationship between climate events and diabetes.[39 43] Three studies examined overweight[40 44] or obesity,[40] two heart disease,[39 43] and one hypertension.[39]

### Food system pathways through which climate change influences nutrition outcomes and DR-NCDs in SSA

Table 2 shows the results on how climate change influences DR-NCDs. The table is organised based on the food system pathways identified from the theoretical literature and presented below. From the six pathways hypothesised, we found evidence of four.

### Food production and retail pathway

Related to food production and retail pathways, four studies examined the relationships between precipitation, temperature and various health outcomes, including obesity,[40] overweight,[40 44] heart disease[45] and DR-NCDs in general.[39]

### Diabetes, hypertension, heart attack, cancer, overweight and obesity risk

Two studies,[39 40] indicated a reduction in food production due to variable rainfall and temperature. Findings showed that as annual precipitation and temperature increase due to climate change, so does the likelihood of women being overweight and obese. This was attributed to global warming, which potentially led to water scarcity in Kenya, reduced availability of nutritious crops and increased reliance on non-nutritious foods.[40] In addition, due to the rising temperatures, rainfall patterns have been unpredictable, causing droughts and famines and thus prompting food insecurity and changes in dietary patterns.[40] Another study attributed the occurrence of diabetes, hypertension, heart attack, cancer and obesity to harsh weather patterns, which disrupted agricultural production, resulting in higher costs of food in urban areas due to reliance on imported produce.[39] Additionally, climate change-driven reliance on chemically treated food production raised concerns about food quality.[39]

**Table 1** Overview of included studies

| Author | Year | Country | Study design | Aspect of climate change | DR-NCDs and risk factors |
|---|---|---|---|---|---|
| Omer et al[46] | 1998 | Sudan | Cross-sectional survey | Temperature (related to increased humidity) | Liver cancer |
| Lontchi-Yimagou et al[43] | 2016 | Cameroon | Retrospective study | Temperature and rainfall | Diabetes |
| Sirma et al[47] | 2019 | Kenya | Cross-sectional survey | Rainfall, droughts, temperature | Liver cancer |
| Odonkor and Sallar[45] | 2020 | Ghana | Descriptive qualitative | Global warming | Heart disease |
| Ngwira et al[44] | 2020 | Malawi | Cross-sectional analysis | Rainfall | Overweight |
| Kasomo and Gayawan[40] | 2021 | Kenya | Cross-sectional analysis | Temperature and rainfall | Overweight, obesity |
| Akello et al[38] | 2021 | Zimbabwe | Cross-sectional analysis | Drought, long dry spells (related to humidity) | Liver cancer |
| Kimanya et al[41] | 2021 | Tanzania | Cross-sectional survey | Temperature (related to increased humidity) | Liver cancer |
| Greibe Andersen et al[39] | 2021 | Kenya | Qualitative | Climate change | Diabetes, hypertension, heart attack and obesity |
| Kortei et al[42] | 2022 | Ghana | Cross-sectional survey | Temperature (related to increased humidity) | Liver cancer |

DR-NCD, diet-related non-communicable diseases.

## Overweight risk

Contrary to other studies, one study suggested that rainfall variability could increase food production, potentially contributing to overweight. The author argued that high rainfall is often associated with better yields fostering overeating behaviours, which may enhance the risk of overweight.[44]

## Heart disease risk

The association of global warming with food production and heart disease was assessed in Ghana. Respondents perceived that heart disease was a result of global warming.[45] As this was based on opinions, the link between climate change and heart diseases was unclear. While climate change implications to heart diseases could be mediated, for example, by heatwaves,[48] our focus was on those impacts mediated via the wider food systems to DR-NCDs in SSA and we could not find sufficient evidence to include influence on heart disease to the HLPE-SSA adapted framework.

## Food storage pathway

Related to food storage pathway, five studies[38 41 42 46 47] examined how temperature (related to increased humidity) and rainfall correlate with liver cancer risks through food storage practices. AF contamination was reported in maize,[38 41] small grains (finger millet, pearl millet, sorghum),[38] milk,[47] peanuts[46] and burkina—a millet-based fermented milk beverage.[42]

## Liver cancer risk

In Zimbabwe, mycotoxins as a result of increased rainfall, heat and drought were associated with liver cancer.[38] The high temperatures were related to humid conditions in West Sudan, which favoured AF growth in peanuts, in a region reported with high liver cancer incidences.[46] A study conducted in four agroecological zones in Kenya found that AF exposure from milk was associated with minimal incidence of liver cancer.[47] Consumption of small grains[38 41 46] and milk consumption posed low liver cancer risks[47] while maize[38] and burkina[42] consumption presented high risks. Peanuts also showed an increased risk of liver cancer, particularly in humid conditions.[46]

## Food consumption pathway
### Diabetes risk

One study investigated the impact of rainfall and temperature on food consumption and its correlation with diabetes.[43] This study revealed that the highest precipitation level coincided with the peak number of newly diagnosed diabetes cases. At the same time, the hospital admission rates for diabetic patients were slightly higher during the rainy season. The authors attributed this phenomenon to nutritional variability. During the long rainy season, physical activity reduces and people tend to consume high-fat and high-calorie food while awaiting the harvest. Consequently, increased caloric

**Table 2** Description of studies included in the review by food system pathway

| Author/year | Country | Study design | Method of analysis | Study objectives | Sample size | Climate change component | Outcome | Key findings |
|---|---|---|---|---|---|---|---|---|
| **Food production pathway** | | | | | | | | |
| Greibe Andersen et al 2021[39] | Kenya | Qualitative | Descriptive statistics | To explore knowledge and perspectives on climate change and health-related issues, with a particular focus on noncommunicable diseases | 28 | Climate change | Diabetes, hypertension, heart attack, cancer and obesity | Climate change-induced food insecurity and rising food prices were considered as prominent challenges. |
| Kasomo and Gayawan 2021[40] | Kenya | Cross-sectional analysis | Inferential statistics | To quantify spatial variations and estimate the effect of climatic and environmental factors on under- and over-nutrition among women in Kenya | 40300 | Temperature and precipitation | Obesity and overweight | Higher precipitation levels were associated with an increased likelihood of women being overweight and obese. |
| Ngwira 2020[44] | Malawi | Cross-sectional analysis | Multivariate probit model | To investigate the effects of climate and location using the multivariate model of malnutrition | 5149 | Rainfall | Overweight | A positive correlation was observed between rainfall and overweight individuals. |
| Odonkor and Sallar 2020 et al[45] | Ghana | Descriptive cross-sectional design | Inferential statistics | To investigate public knowledge of global warming and its effects on human health | 1130 | Global warming | Heart disease | Rising food prices impact the affordability and availability of food in urban areas. |
| **Food storage pathway** | | | | | | | | |
| Akello 2021 et al[38] | Zimbabwe | Cross-sectional analysis | Inferential statistics | To assess frequencies of mycotoxin-producing fungi in maize, sorghum, pearl millet and finger millet cropped and traded in Zimbabwe | NI | Drought, long dry spells, | Liver cancer | Consumption of aflatoxin-contaminated maize stored for 4–6 months increased liver cancer rates. |
| Kimanya et al 2021[41] | Tanzania | Cross-sectional analysis | Inferential statistics | To estimate the population risk of aflatoxin-induced liver cancer | NI | Temperature | Liver cancer | Population risk for aflatoxin-induced liver cancer nationwide was estimated to be 2.95 cases per 100000 people. |
| Kortei et al 2022[42] | Ghana | Cross-sectional survey | NI | To evaluate $AFM_1$ levels and cancer risks associated with burkina (a millet-based fermented milk product) | 150 | Temperature | Liver cancer | Consumption of burkina showed varied risk levels of liver cancer. |
| Omer et al 1998[46] | Sudan | Cross-sectional survey | Inferential statistics | To investigate whether aflatoxin contamination of peanut products may contribute to the incidence of hepatocellular carcinoma in Sudan | 54 | Temperature | Liver cancer | Aflatoxin contamination of peanuts has a significant association with liver cancer. |

Continued

**Table 2** Continued

| Author/year | Country | Study design | Method of analysis | Study objectives | Sample size | Climate change component | Outcome | Key findings |
|---|---|---|---|---|---|---|---|---|
| Sirma et a 2019[47] | Kenya | Cross-sectional survey | Inferential statistics | To assess the risk of liver cancer posed by $AFM_1$ in milk, assuming 10-fold lower carcinogenicity than $AFB_1$ | NI | Rainfall, droughts, temperature | Liver cancer | Aflatoxin exposure from milk contributed a small percentage to liver cancer incidence rates. |
| Food consumption pathway | | | | | | | | |
| Lontchi-Yimagou et al 2016[43] | Cameroon | Retrospective study | Inferential statistics | To investigate the relationship between diabetes hospitalisation rates and climate variations in Yaoundé | 818 | Temperature and precipitation | Diabetes | Diabetes admissions were higher during rainy seasons due to nutritional variability. |

NI, not indicated.

intake could lead to hyperglycaemia and the onset of diabetes symptoms.[43]

## Grey literature findings

The majority of the reports from the grey literature lacked specificity of region and focus. Among the few reports that explored the relationship between climate change, food systems and DR-NCDs, they did not provide country-specific results[18] or did not explicitly describe which health outcome[49] was being addressed. Although most reports mentioned how climate change impacted food security and malnutrition, the studies did not mention which type of malnutrition (undernutrition or DR-NCDs) it was focusing on.[49] Thus, they were not included in this review. We explored a sample of governments but could not find relevant studies to include in this review.

## Conceptual framework of food systems for DR-NCDs and nutrition with evidence drawn from the local context

Figure 3 presents an amended food systems conceptual framework, named here HLPE-SSA, for understanding the pathways and mechanisms in which climate change impacts on food systems and DR-NCDs in SSA. HLPE-SSA is amended from the HLPE framework and is based on the evidence and results discussed above. While both the HLPE and HLPE-SSA show the pathways in which climate change is linked to food systems and DR-NCD outcomes, HLPE-SSA differs slightly from HLPE. *First* is that HLPE-SSA provides a region-specific analysis of how climate change impacts food systems and DR-NCDs. Due to limited evidence from SSA, we expect the HLPE-SSA framework to change with further data from other areas and new research, for example, with respect to heart disease. *Second*, while HLPE shows more generalised linkages between climate change and nutrition outcomes, HLPE-SSA provides a more in-depth synthesis, specifying the mechanisms by which climate change affects food systems and nutrition. For instance, HLPE-SSA shows how temperature can disrupt agricultural production, reducing food availability and subsequent impacts on the risk of diabetes. *Third*, HLPE-SSA proposes a potential pathway—the food consumption pathway— which is not present in the HLPE framework. This pathway acknowledges that diets and unhealthy consumption practices could also be mediated by other factors beyond food production, such as mobility, marketing, retail and trade policies.

The evidence from HLPE-SSA aligns with four (production, storage, retail and consumption) of the six hypothesised pathways.

Regarding the food production pathway, HLPE-SSA confirms our hypothesis. Variations in temperature and precipitation influence food production. Consequently, these changes impact on the quality and quantity of food, which were associated with the risk of diabetes, hypertension, heart attack, obesity and overweight.

For the food storage pathway, HLPE-SSA confirms our hypothesis regarding the effect of long dry spells, drought

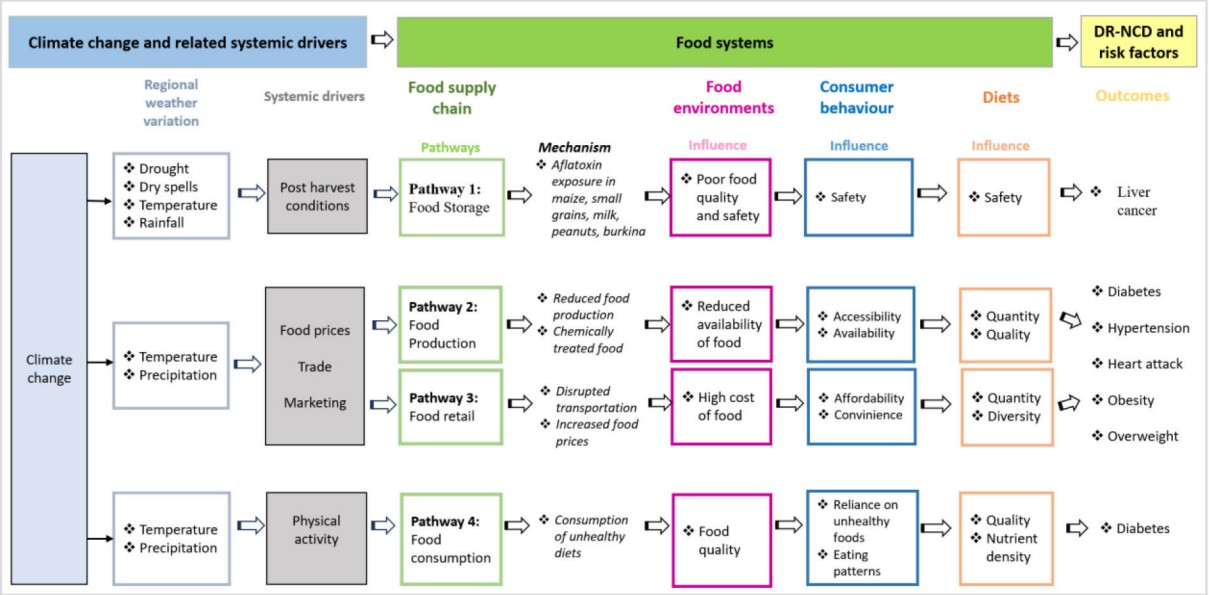

**Figure 3** Food systems conceptual framework for sub-Saharan Africa: HLPE-SSA. Figure adapted from HLPE. 2017 [18]. Arrows show the expected causal effect. HLPE, high-level panel of experts; DR-NCDs, diet related non-communicable diseases; SSA, sub-Saharan Africa.

and fluctuations in rainfall and temperature on food storage and, in turn, liver cancer in SSA. The impact was more pronounced in poor storage and harvesting conditions of grains, whereas unfavourable weather conditions, such as high temperatures and rainfall, increased the risk. These weather conditions impacted humidity levels (above 14% moisture content) increasing AF growth in grains and milk. AF exposure in these grains and milk was linked to the risk of liver cancer.

Regarding the food retail pathway, HLPE-SSA suggests that variations in temperature and rainfall patterns affected trade and markets, disrupting transportation, and increasing food prices. This, in turn, increased the cost of food, thereby impacting its affordability and convenience. These changes in the food systems were associated with an increased risk of diabetes, hypertension, heart attack, obesity and overweight.

Regarding the food consumption pathway, HLPE-SSA supported our hypothesis that temperature and rainfall patterns alter eating patterns and physical activity during the long rains. This resulted in reduced food quality and nutrient density, which was associated with high diabetes risk.

Lastly, HLPE-SSA framework highlights research gaps related to the food distribution, market and food processing and packaging pathways. Although we hypothesised that extreme weather events might disrupt the distribution and processing of food in SSA—and consequently on DR-NCDs. To the best of our knowledge, no studies investigated how climatic impacts can influence DR-NCDs through these two pathways.

## DISCUSSION

This scoping review focused on studies on the SSA context to better understand the relevance of local context to food systems and DR-NCDs. It indicates very limited research focus and calls for further research on food systems and DR-NCDs in SSA. Despite this, findings from our review supported our hypotheses that climate change influences food systems through food production, food storage, food retail and food consumption pathways. These changes in the food chains compromise food quality, safety, quantity and nutrient density. These factors contribute to a shift towards less nutritious and more processed foods associated with developing DR-NCDs.

Food storage pathway focuses on how climate-induced changes in rainfall and precipitation affect food harvest and storage conditions, leading to the growth of AF $B_1$ and AF $M_1$ ($AFM_1$)—a type of mycotoxin produced by fungi *Aspergillus flavus* and *Aspergillus parasiticus*—present in grains, maize, nuts, peanuts, flour and milk.[50] The review's findings are consistent with other research conducted in Bangladesh, which reported a significant proportion of liver cancer cases attributed to dietary AF exposure from maize and other crops in warm and humid conditions.[51] Notably, based on the findings of this review, the risk of AF contamination and its associated liver cancer risk was found to present a lower risk in milk and other grains. This observation is in line with studies by Saha Turna *et al* conducted in Bangladesh[52] and globally,[51] where AF exposure in milk and grains such as wheat and rice was not associated with liver cancer. Several factors are identified as influencing AF contamination, including storage conditions, duration and type of crop storage structure. AFs are prone to develop during food production in dry and humid conditions, whereas wet and humid conditions

during harvesting and storage create conducive conditions for the growth of AFs.[53]

Related to food production pathway, evidence from Cameroon, Kenya and Malawi showed how climate variables correlate with diabetes, heart disease and obesity. Seasonal changes in rainfall were found to influence physical activity, eating habits and dietary choices, and in turn diabetes and obesity in the region. The review's findings align with other studies that have linked disruptions in agricultural production resulting from extreme weather events with diabetes and obesity.[54] During rainy season and extreme weather events, agricultural production activities are disrupted, leading to food shortages.[42] During food shortages, individuals with diabetes may not consume sufficient fruit and vegetables, and compensate with increased energy-dense, high-fat/salt foods, worsening their condition.[42] Another plausible reason could be that people would be more likely to seek care during the rainy season as they are not busy with farm activities.

The increased access to and consumption of processed and unhealthy foods may contribute to the burden of DR-NCDs. Food consumption pathway is supported by a review[27] highlighting that extreme weather events are likely to become more frequent determinants of food purchase and consumption. These events can have a dual impact, either by restricting consumers' access to food or by shaping their food preferences based on the availability of certain product.[27]

Evidence from SSA supported four of the hypothesised pathways. Studies in this review did not support the food distribution, market and processing and packaging pathways. This could be attributed to the limited evidence and studies included in the review and hence could not support all the pathways.

In summary, the HLPE-SSA framework is based on very limited research, but it provides a structured approach to understanding the complex interactions between climate change and related systemic risk, food systems and DR-NCDs in SSA. It is important that our amendments to the HLPE framework are considered as the first exploratory findings calling for further research from SSA in the area for more comprehensive context-based understanding.

## Strengths

This review exhibits several notable strengths.

First, the HLPE framework has been extensively used by other researchers,[55 56] which strengthens the methodology of this review.

Second, this review used an established food systems framework to guide the methodology and research process. This provided a clear structure to clarify pathways on the causal relationship between climate change, food systems and DR-NCDs.

Third, the review goes beyond existing frameworks by proposing a context-specific framework tailored to SSA. Similar frameworks for food systems exist, but they are extensive in scope. HLPE-SSA, offers an in-depth

and specific synthesis of how climate change and certain systemic drivers influence SSA's food systems and contribute to the development of DR-NCDs. This approach enhances the framework's applicability in guiding targeted interventions such as better crop storage, sustainable agriculture and policies that address the region's specific concerns.

Lastly, the review adhered to a systematic approach, following JBI guidance for conducting scoping reviews. This systematic process ensures the reliability and comprehensiveness of the research process.

## Limitations

One key limitation of the review is the possibility of publication bias, as the review focused on international organisations and academic publications. Relevant studies published in country-based databases as well as local languages, might have been missed. However, our search of a sample of government websites did not indicate that we missed substantial body of literature. In addition, even though we did not restrict our search to English, most articles retrieved from our search were in English and thus this might have biased the results.

Additionally, the nature of the studies included in the review—mainly observational and descriptive—limits the establishment of definitive causal relationships between climate change, food systems and DR-NCDs. Although they offered insights and generated hypotheses of the potential linkages, they may not provide conclusive evidence on causality. Further research, including natural experiments, longitudinal and time series analysis studies, would be necessary to establish more causal links. This research would also need to focus on the proportional role of different pathways of food systems in relation to DR-NCDs in changing societies, dietary and consumption practices in SSA.

Some studies, mainly qualitative studies, looked at the linkages between climate change and DR-NCDs without showing how they influence each other. Therefore, we had to assemble the scattered evidence from these studies to form a comprehendible picture of these linkages. However, these linkages were not included in the HLPE-SSA framework.

Furthermore, this review did not include studies exploring the potential effects of floods and famine on DR-NCDs. Floods can impact food security by destroying crops and sometimes leading to famine. Floods also destroy the socioeconomic activities linked to food production, which could impact food accessibility, affordability and utilisation. On the other hand, empirical studies have shown that famine exposure in the first 1000 days has been associated with higher risk of diabetes in adulthood[57 58] (fetal famine), which represents another area that could be relevant to the overall understanding of climate change impacts on DR-NCDs in SSA. Including such evidence in future research could contribute to a more comprehensive analysis of the factors influencing DR-NCDs in the region.

## Conclusion

In conclusion, this review suggests climate change can significantly affect food systems, nutrition and DR-NCDs in SSA. It highlights the association between AF-contaminated food and liver cancer in multiple countries. It also underscores how seasonal variations in rainfall can influence physical activity and dietary choices, potentially exacerbating diabetes. Furthermore, extreme weather events are identified as influential factors shaping food purchase and consumption patterns, potentially limiting access to nutritious foods and influencing food preferences.

Our work on the HLPE-SSA suggests that while wider conceptual frameworks such as HLPE work as a general reference, it is important for research and practice to cast light also to regional adjustments of more global frameworks. This review further highlights the key areas for further work such as the mechanism in which climate change influence heart diseases and the gaps related the food distribution, market and processing and packaging pathways.

**Author affiliations**
[1]Department of Health Sciences, Faculty of Social Sciences, Tampere University, Tampere, Finland
[2]Department of Population Health Sciences, School of Life Course & Population Sciences, Kings College London, London, UK
[3]Department of Global Public Health, Karolinska Institute, Stockholm, Sweden
[4]WHO Collaborating Centre on Health in All Policies and the Social Determinants of Health, Health Sciences, Faculty of Social Sciences, Tampere University, Tampere, Finland

**Contributors** JT, SH, JP and MK conceptualised the article. JT and MK conducted the screening of studies. JT did data extraction. JT, SA, JP, SH and MK contributed to the study methodology. JT, SH, MK, SA and JP contributed to writing–review and editing. JT contributed to writing–original draft. JT responsible for the overall content as guarantor.

**Funding** JT was supported by the EDUFI Fellowship (Finnish National Agency for Education). SH was funded by the Medical Research Council grant number MR/S009035/1, MR/N015959/1, MR/S003444/1, MR/Y009983/1, MR/X009777/ and MR/X003078/1. MK, JP and SA have not declared grant from any funding agency in the public, commercial or not-for-profit sectors.

**Competing interests** None declared.

**Patient and public involvement** Patients and/or the public were not involved in the design, or conduct, or reporting, or dissemination plans of this research.

**Patient consent for publication** Not applicable.

**Provenance and peer review** Not commissioned; externally peer reviewed.

**Data availability statement** All data relevant to the study are included in the article or uploaded as online supplemental information.

**ORCID iDs**
Janet Tapkigen http://orcid.org/0000-0002-9738-4965
Meri Koivusalo http://orcid.org/0000-0002-5365-5372

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
