## [Reviewer comments · BMJ Open]

ARTICLE DETAILS

TITLE (PROVISIONAL)	Climate Change-induced Shifts in Food Systems and Diet-related Noncommunicable Diseases in Sub-Saharan Africa: A Scoping Review and a Conceptual Framework
AUTHORS	Tapkigen, Janet; Harding, Seeromanie; Jutta, Pulkki; Atkins, Salla; Koivusalo, Meri

VERSION 1 – REVIEW

REVIEWER	Ali, Zakari Medical Research Council Unit The Gambia at the London School of Hygiene and Tropical Medicine, Nutrition and Planetary Health
REVIEW RETURNED	27-Oct-2023

GENERAL COMMENTS	Dear authors, thank you for the opportunity to read your study. As someone who has been monitoring developments in this field, I think this review is very much needed and it summarises the evidence to date and highlights the key areas for further work. I really think this study addresses a key topic at the right time and have some comments below for author consideration. - I think it is important to adapt the HLPE global framework for DR-NCDs in SSA as it helps to clarify the pathways in a very complex causal relationship. I am only a bit concerned that your adapted version does not look particularly different from the original HLPE framework or its interpretations.- The difference between climate change and unfavourable weather events on DR-NCDs. How does a study on the effects of humidity on liver cancer be regarded as the impact of climate on liver cancer? What level of humidity change over how many years is climate change?- What is the difference between the January and December database searches? Was the search repeated, and why?- Page 10, lines 3-4: "We included studies published in peer-reviewed journals between 1995 and 2022." What did you do with the grey literature search then? Needs clarification.
---

REVIEWER	Charnley, Gina Imperial College London Faculty of Medicine, School of Public Health, Department of Infectious Disease Epidemiology
REVIEW RETURNED	18-Dec-2023

GENERAL COMMENTS	Thank you for the opportunity to review the manuscript entitled "Climate change-induced Shifts in Food Systems and Diet-related
---

Noncommunicable Diseases in Sub-Saharan Africa: A Scoping Review and Conceptual Framework". I think this is a very interesting topic and relatively unexplored. The paper is very well written and the aims are clear. Take care with your line numbers, they re-start on each page, which makes reviewer feedback tricky.

Some more detailed feedback relating to each section is below.

Summary

Line 10: I think I would remove "High-Level Panel of Experts" and just say "while it has been provided that..." Abstract/summaries should be stand-alone and its impossible to know what this means without reading the manuscript.

Line 19-20: Incomplete sentence "...to disprove it?", disprove what?

"Implications of all available evidence", I have not seen the submission guidelines for BMJ Open, and so perhaps this section is correct. However, as I understand it, this isn't implications of "all evidence", just the results here and the HLPE framework. There is clearly more evidence than that, as your review included 10 studies.

Introduction

Line 57: I think changing behaviour/patterns of extreme weather is better than increasing frequency, some extreme weather events are not increasing, but instead changing in intensity and occurring out of season (<https://www.nature.com/articles/s41558-022-01388-4>).

Methods

Line 53: Briefly state what you mean by the "food production pathway"

I see, these now refer to the pathways in the first paragraph, perhaps consider restricting this so it says, "These include: (next paragraph) 1, The food production pathway, (next paragraph) 2, The food storage pathway, etc

Line 6, page 8: What is a "conductive environment"?

Line 39: Not sure JBI was spelling out on first use

Line 9, page 9: Not sure you need number 3, I would image it will be written in some language.

Line 46: Why the first 20 pages of Google Scholar?

Line 58: I do not see "Appendix 1" in this manuscript, it would have been useful to see your search criteria, particularly how you search for "climate change".

Line 40, page 10: This is not a role of funding statement.

Results

Why do you discuss 6 pathways in your "Conceptual framework development" section of the methods, and then only have paragraph for three of them at the end of your results and in the discussion?

Line 12, page 13: "climate vents"?

Figure 3: This is exactly the same as Figure 1, with some sections slightly re-worded and with the addition on some shading. So I am not sure what this adds? Perhaps if this is not the case, a more detailed figure caption might help to explain what this adds, that Figure 1 does not.

	Line 24, page 16: “dried, hotter regions”, which regions are these? Line 5-55 page 16: I think you can probably condense and better summaries the “Food storage pathway” section, its a lot of words making a similar point. Line 13, page 17: How did they state that rainfall variability increase food production? Line 18: Again, how does global warming increase heart disease? Surely the confounders would make trying to link that near impossible? Discussion Line 5-8, page 18: The review does not really indicate that extreme weather such as floods and droughts, can disrupt agricultural productivity, you don’t mention floods at all in your results. Not sure I understand why there needs to be a sub-heading for “Conceptual framework”? in the discussion. I think throughout better consistency of sub-headings through methods, results and discussion would make your review easier to follow. Line 13-17, page 19: You talk about an “established framework”, but you never actually explain this framework, and I think at the moment, its a major limitation of the manuscript. The “High-Level Panel of Experts” is something discussed in your abstract and summaries at the beginning, you then go back to mentioning in the methods, and it appears fundamental to the framework you are building your review on. However, nowhere in the introduction do you mention the framework, or the HLPE (e.g., who are they, what are they experts in, when were they formed, what are their aims?). So this makes understanding how you arrived on this framework and methods very challenging. Line 20-23: You do not provide evidence in your review for this, I cant see how Figure 3 is different to Figure 1. Line 36, page 19: You stating that you were not excluding based on language, but I assume you only searched in one language, so that biased your results, not just excluding local languages, but all languages that were not in the one you used. Line 48: What is “foetal famine”? Conclusion Also include some concluding remarks of what you found, not just what you didn’t find or what still needs to be done.
--	--

VERSION 1 – AUTHOR RESPONSE

Response to Reviewer 1

Dear authors, thank you for the opportunity to read your study. As someone who has been monitoring developments in this field, I think this review is very much needed and it summarises the evidence to date and highlights the key areas for further work. I really think this study addresses a key topic at the right time and have some comments below for author consideration.

Response: Thank you for your thoughtful comments. We are pleased to hear that the review is valuable in addressing the current key topic and state of the art. The responses to the questions are as follows.

1. I think it is important to adapt the HLPE global framework for DR-NCDs in SSA as it helps to clarify the pathways in a very complex causal relationship. I am only a bit concerned that your adapted version does not look particularly different from the original HLPE framework or its interpretations.

Response: Thank you for your comment. We have now used the original HLPE framework (with permission from the HLPE copyrights team) in the methods to hypothesise the causal pathways; see Figure 1. HLPE-SSA has also been redrawn to better describe the evidence found in this review; see Figure 3.

2. The difference between climate change and unfavourable weather events on DR-NCDs. How does a study on the effects of humidity on liver cancer be regarded as the impact of climate on liver cancer? What level of humidity change over how many years is climate change?

Response: Thank you for your comment. Previously, we had classified the climate variable as humidity if a study examined how rising temperature leads to increased humidity and, in turn, liver cancer. To clarify this, we have renamed the climate variable to "temperature (related to increased humidity)." See page 11, line 320-324 and page 13, lines 363-364; as well as Table 1 and Table 2.

3. What is the difference between the January and December database searches? Was the search repeated, and why?

Response: Thank you for your comment. The January database search was a pilot search to test the search terms, while the December database search was the main search and the one utilised in this review. We have now removed the reporting of the pilot search from the methods; see page 9, lines 251-256.

4. Page 10, lines 3-4: "We included studies published in peer-reviewed- journals between 1995 and 2022." What did you do with the grey literature search then? Needs clarification.

Response: Thank you for your comment. Grey literature was searched for, but we did not report these findings as we did not find relevant material for the review. However, we have now added the sections on grey literature search and reporting in the methods section, page 9, lines 247-249 and pages 9-10, lines 257-272.

Response to Reviewer 2

We appreciate your positive feedback regarding our article. We are pleased that you found the review interesting and consider it relatively unexplored. Below are the responses to the questions.

1. Thank you for the opportunity to review the manuscript entitled "Climate change-induced Shifts in Food Systems and Diet-related Noncommunicable Diseases in Sub-Saharan Africa: A Scoping Review and Conceptual Framework". I think this is a very interesting topic and relatively unexplored. The paper is very well written, and the aims are clear. Take care with your line numbers, they re-start on each page, which makes reviewer feedback tricky.

Response: Thank you for your comments. We acknowledge our mistake regarding the line numbers and apologise for the inconvenience it may have caused during the review. We have addressed this issue in the revised manuscript.

Summary

2. Line 10: I think I would remove “High-Level Panel of Experts” and just say “while it has been provided that...” Abstract/summaries should be stand-alone and its impossible to know what this means without reading the manuscript.

Response: Thank you for your comment. This section has been corrected on page 2, lines 36-39.

3. Line 19-20: Incomplete sentence “..to disprove it?”, disprove what?

Response: Thank you for your comment. This statement has been removed and rewritten; see page 2, lines 36-39.

4. “Implications of all available evidence”, I have not seen the submission guidelines for BMJ Open, and so perhaps this section is correct. However, as I understand it, this isn’t implications of “all evidence”, just the results here and the HLPE framework. There is clearly more evidence than that, as your review included 10 studies.

Response: Thank you for your comment and for flagging this. The guidelines for BMJ Open do not have the "Implications of all available evidence" section. We have removed this section and added a section entitled 'Strengths and limitations of this study' to match the guidelines on page 3, lines 50-61.

Introduction

5. Line 57: I think changing behaviour/patterns of extreme weather is better than increasing frequency, some extreme weather events are not increasing, but instead changing in intensity and occurring out of season (<https://www.nature.com/articles/s41558-022-01388-4>).

Response: Thank you for your comment. We have replaced this with ‘changing patterns of extreme weather events’ on page 4, line 101.

Methods

6. Line 53: Briefly state what you mean by the “food production pathway.”

I see, these now refer to the pathways in the first paragraph, perhaps consider restricting this so it says, “These include: (next paragraph) 1, The food production pathway, (next paragraph) 2, The food storage pathway....., etc

Response: Thank you for your comment. We have now rewritten this section on pages 7-7, lines 161-196, as advised.

7. Line 6, page 8: What is a “conductive environment”?

Response: Thank you for your comment. We apologise for the typing error. We have changed this to "conductive environment" on page 7, line 167.

8. Line 39: Not sure JBI was spelling out on first use.

Response: Thank you for your comment. We have now spelt it out on page 8, line 201.

9. Line 9, page 9: Not sure you need number 3, I would image it will be written in some language.

Response: Thank you for your comment. In our database search, we did not restrict the search to English or any other language; see line 218. Thus, we have acknowledged this as a limitation of the study.

10. Line 46: Why the first 20 pages of Google Scholar?

Response: Thank you for your comment. The 20 pages is an often-used reference for research reviews, but this is not necessarily sufficient for grey material. We have now followed Haddaway et al.'s 2015 suggestions to include the first 80 pages to gain further reach for grey literature. See page 9, lines 257-260.

11. Line 58: I do not see "Appendix 1" in this manuscript, it would have been useful to see your search criteria, particularly how you search for "climate change".

Response: Thank you for your comment. The search terms for academic and grey literature have been added as Appendix 1 and Appendix 2.

12. Line 40, page 10: This is not a role of funding statement.

Response: Thank you for your comment. We have added the funding statement following the BMJ Open guidelines on page 25, lines 714-717.

Results

13. Why do you discuss 6 pathways in your "Conceptual framework development" section of the methods, and did not at the end of your results and in the discussion?

Response: Thank you for your comment. We have now discussed this in the results on pages 16-17, lines 415-441, and at the end of the discussion on page 19, lines 484-492.

14. Line 12, page 13: "climate vents"?

Response: Thank you for your comment. We apologise for the typing error. This has been corrected on page 12, line 327.

15. Figure 3: This is exactly the same as Figure 1, with some sections slightly re-worded and with the addition on some shading. So, I am not sure what this adds. Perhaps if this is not the case, a more detailed figure caption might help to explain what this adds, that Figure 1 does not.

Response: Thank you for your comments. The framework has now been redone. Figure 1 is now the original HLPE framework (with copyright permission from the HLPE team, and Figure 3 is structured from the evidence of this review.

16. Line 24, page 16: "dried, hotter regions", which regions are these?

Response: Many thanks for your comment. This section has been rewritten in lines 362-375.

17. Line 5-55 page 16: I think you can probably condense and better summaries the "Food storage pathway" section, it's a lot of words making a similar point.

Response: Thank you for your comment. This has been shortened and restructured. See pages 13, lines 362-375.

18. Line 13, page 17: How did they state that rainfall variability increases food production?

Response: Thank you for your comment. This has been addressed on page 12, lines 339-350.

19. Line 18: Again, how does global warming increase heart disease? Surely the confounders would make trying to link that near impossible?

Response: Thank you for your comment. Climate change implications for heart diseases could be mediated, for example, by heat waves. However, the focus of our review was not any impacts of climate change on NCDs but those that are mediated via the wider food systems to diet-related NCDs, with a focus on studies originating from SSA country contexts. We have rewritten this paragraph; see page 12, lines 355-361.

Discussion

20. Line 5-8, page 18: The review does not really indicate that extreme weather such as floods and droughts, can disrupt agricultural productivity, you don't mention floods at all in your results.

Response: Thank you for your comment. This was an error. We have removed floods from the text. See lines 444-451.

21. Not sure I understand why there needs to be a sub-heading for "Conceptual framework"? in the discussion. I think throughout better consistency of sub-headings through methods, results and discussion would make your review easier to follow.

Response: Thank you for your comment. This sub-heading has been removed.

22. Line 13-17, page 19: You talk about an "established framework", but you never actually explain this framework, and I think at the moment, its a major limitation of the manuscript. The "High-Level Panel of Experts" is something discussed in your abstract and summaries at the beginning, you then go back to mentioning in the methods, and it appears fundamental to the framework you are building your review on. However, nowhere in the introduction do you mention the framework, or the HLPE (e.g., who are they, what are they experts in, when were they formed, what are their aims?). So, this makes understanding how you arrived at this framework and methods very challenging.

Response: Thank you for your comment. The description of HLPE has now been incorporated into the introduction on page 5, lines 128-136.

23. Line 20-23: You do not provide evidence in your review for this, I can't see how Figure 3 is different to Figure 1.

Response: Thank you for your comment. The two frameworks have now been modified. Figure 1 is now the original HLPE framework, and Figure 3 is a modified HLPE now called HLPE-SSA (based on the findings from this review).

24. Line 36, page 19: You are stating that you were not excluding based on language, but I assume you only searched in one language, so that biased your results, not just excluding local languages, but all languages that were not in the one you used.

Response: Thank you for your comment. Yes, this is true. We have now added this as a limitation of the study in lines 515-516.

25. Line 48: What is “foetal famine”?

Response: Thank you for your comment. This implies foetal famine exposure and refers to a situation where the developing foetus experiences inadequate nutrient supply during pregnancy due to famine with consequences on NCDs later in life. This section has been rewritten for clarity on page 20, lines 532-536.

Conclusion

26. Also include some concluding remarks of what you found, not just what you didn’t find or what still needs to be done.

Response: Thank you for your comment. We have included conclusions on the study findings and what needs to be done; see page 20, lines 539-550.

VERSION 2 – REVIEW

REVIEWER	Charnley, Gina Imperial College London Faculty of Medicine, School of Public Health, Department of Infectious Disease Epidemiology
REVIEW RETURNED	18-Mar-2024

GENERAL COMMENTS	Thank you for submitting your revisions, which were carefully addressed and clearly answered. I believe this has really strengthened your contribution and made your important findings and conclusions much easier to interpret. I have one further comment, regarding a previous comment which I do not feel was sufficiently answered. Inclusion criteria 3 and previous revision comment 24: Including a paper because it was "written in any language" is not an inclusion criteria. As all papers are written in a language, and all papers would fall into this criteria, making it redundant. You state that no language restrictions were implied to the search results, which is great. However, unless you searched in every possible language, then not all languages will be included, only the ones that had an abstract or title in the language which you searched. You need to modify your inclusion criteria to represent this more clearly. E.g., "Inclusion criteria 3. Studies which had an English
---

	abstract or title". If English was the only language you searched in? It looks like from Appendix 1, it was only English. As it is, I could not replicate your study, as you do not state all the languages you searched in, which will massively bias your results. However, this is a bias of all reviews, as it is not realistic to search in every possible language. You just need to make it clear. Thank you for your continued effort in improving your manuscript.
--	--

VERSION 2 – AUTHOR RESPONSE

Reviewer’s comment: Thank you for submitting your revisions, which were carefully addressed and clearly answered. I believe this has really strengthened your contribution and made your important findings and conclusions much easier to interpret. I have one further comment, regarding a previous comment which I do not feel was sufficiently answered.

Response: Thank you for your feedback and appreciate the opportunity to improve the overall understanding and replication of our review. We are pleased to hear that our previous revisions strengthened the understanding of our findings and conclusions.

Reviewer’s comment: Inclusion criteria 3 and previous revision comment 24: Including a paper because it was "written in any language" is not an inclusion criteria. As all papers are written in a language, and all papers would fall into this criteria, making it redundant.

You state that no language restrictions were implied to the search results, which is great.

However, unless you searched in every possible language, then not all languages will be included, only the ones that had an abstract or title in the language which you searched.

You need to modify your inclusion criteria to represent this more clearly. E.g., "Inclusion criteria 3. Studies which had an English abstract or title". If English was the only language you searched in? It looks like from Appendix 1, it was only English.

As it is, I could not replicate your study, as you do not state all the languages you searched in, which will massively bias your results. However, this is a bias of all reviews, as it is not realistic to search in every possible language. You just need to make it clear.

Response to comments: Our search was conducted in English which may have by default of the search engine returned mostly abstracts in English. We have revised the inclusion criteria and have adjusted it accordingly. Inclusion criteria 3 now states ‘Had an English abstract or title’. See line 215, pg. 8.